# A novel binary pesticidal protein from *Chryseobacterium arthrosphaerae* controls western corn rootworm by a different mode of action to existing commercial pesticidal proteins

Rania Jabeur[1], Virginie Guyon[2], Szabolcs Toth[3,4], Adriano E. Pereira[5], Man P. Huynh[5], Zakia Selmani[6], Erin Boland[7], Mickael Bosio[1], Laurent Beuf[1], Pete Clark[7], David Vallenet[8], Wafa Achouak[9], Carine Audiffrin[10], François Torney[7], Wyatt Paul[1], Thierry Heulin[9], Bruce E. Hibbard[11], Stefan Toepfer[4], Christophe Sallaud[1]*

1 Limagrain Europe, Centre de recherche, Chappes, France, 2 Vilmorin & Cie, Saint-Beauzire, France, 3 Integrated Pest Management Department, Hungarian University of Agriculture and Life Sciences—MATE, Godollo, Hungary, 4 CABI Switzerland, c/o Plant Protection and Soil Conservation Directorate, Hodmezovasarhely, Hungary, 5 Division of Plant Science & Technology, University of Missouri, Columbia, MO, United States of America, 6 Laboratoire de Biologie et Physiologie des Organismes, Faculté des Sciences Biologiques, Université des Sciences et de la Technologie Houari Boumediène, USTHB, Alger, Algérie, 7 Genective USA Corp, Champaign, IL, United States of America, 8 LABGeM, Génomique Métabolique, CEA, Genoscope, Institut François Jacob, Université d'Evry, Université Paris-Saclay, CNRS, Evry, France, 9 Aix Marseille Univ, CEA, CNRS, BIAM, LEMIRE, Saint Paul-Lez-Durance, France, 10 Protéus, Nîmes, France, 11 USDA-ARS, Plant Genetics Research Unit, Univ. Missouri, Columbia, MO, United States of America

* christophe.sallaud@limagrain.com

**Data Availability Statement:** All relevant data are within the manuscript and its Supporting Information files.

## Abstract

The western corn rootworm (WCR) *Diabrotica virgifera virgifera* (Coleoptera: Chrysomelidae) remains one of the economically most important pests of maize (*Zea mays*) due to its adaptive capabilities to pest management options. This includes the ability to develop resistance to some of the commercial pesticidal proteins originating from different strains of *Bacillus thuringiensis*. Although urgently needed, the discovery of new, environmentally safe agents with new modes of action is a challenge. In this study we report the discovery of a new family of binary pesticidal proteins isolated from several *Chryseobacterium* species. These novel binary proteins, referred to as GDI0005A and GDI0006A, produced as recombinant proteins, prevent growth and increase mortality of WCR larvae, as does the bacteria. These effects were found both in susceptible and resistant WCR colonies to Cry3Bb1 and Cry34Ab1/Cry35Ab1 (reassigned Gpp34Ab1/Tpp35Ab1). This suggests GDI0005A and GDI0006A may not share the same binding sites as those commercially deployed proteins and thereby possess a new mode of action. This paves the way towards the development of novel biological or biotechnological management solutions urgently needed against rootworms.

**Funding:** RJ, VG, ST, AP, AMH, ZS, EB, MB, LB, PC, DV, WA, CA, FT, WP, TH, BH, ST, CS Grant number = "Fond Diabrotica" Funder = French Interprofessional Organization for Seeds and Plants" former GNIS association under the call named URL of the funder = https://www.gnis.fr/ The funders had no role in study design, data collection and analysis, decision to publish, or preparation of the manuscript.

**Competing interests:** The authors have declared that no competing interests exist.

## Introduction

Worldwide, maize (*Zea mays* L.) is one of the most extensively grown cereal crops providing a large proportion of carbohydrates for livestock, humans and biofuel [1]. However, maize production is constrained by extreme climatic events, a number of insect pests and a few important diseases [2–5]. The western corn rootworm (WCR) *Diabrotica virgifera virgifera* (LeConte, 1868) (Coleoptera: Chrysomelidae), is amongst the most serious insect pests of maize in the USA and Europe [6, 7]. This pest species is hypothesized to have originated from Mexico or Central America but has invaded all major maize production areas of North America as well as most of Central Europe. Both larvae and adults are responsible for management costs and yield losses of more than $1 billion in the USA each year [6, 8, 9] and expected to account for at least ½ billion dollars in Europe in the near future [10]. Most crop losses from WCR are due to larval feeding on maize roots leading to plant lodging [6]. Crop rotation or the application of insecticides were the major approaches to mitigate rootworm damage before the commercialization of genetically modified maize in the USA in 2003 [11, 12].

During the last decade, pesticidal proteins derived from the bacterium *Bacillus thuringiensis* (*Bt*) Berliner have been successfully used to control rootworms [13] through expression in maize roots. In addition to better yield, Bt maize has resulted in less damage to the environment and human health (farmers and consumers as well) by reducing insecticide use [14, 15]. Despite their use against insects, some Bt strains have proven efficacy in the degradation of insecticides [16].

The first *Bt* maize commercialized against maize rootworm larvae produced the *Bt*-derived protein Cry3Bb1 (MON863 event in 2003) [17]. Subsequently several pesticidal proteins have been used either singly or as pyramids, including Cry34Ab1/Cry35Ab1 (reassigned as Gpp34Ab1/Tpp35Ab1 by the Bacterial Pesticidal Protein Resource Center [BPPRC]) [18], mCry3A, eCry3.1Ab and Cry3Bb1 pesticidal proteins [19–22]. Nevertheless, despite some success of the use of *Bt* maize for managing maize rootworm, field-evolved resistance has been detected against the currently used commercially available pesticidal proteins of either the Cry3A/B or Gpp34Ab1/Tpp35Ab1 family type. The appearance of resistant rootworms to Cry3Bb1 [23], which are cross-resistant to mCry3A [24], and the emergence of Gpp34Ab1/Tpp35Ab1 resistant populations [25], have shown that rootworms can relatively quickly adapt to the modes of action of these pesticidal proteins. Also resistance management strategies like "refuge in the bag" may not be always sufficient to delay the appearance of resistance [26, 27].

Thus, finding agents or ingredients with new modes of action (MoA) against rootworms is the current challenge for the plant protection and maize breeding industry. New approaches involving RNA interference targeting WCR genes were recently tested [28]. For example, *in planta* expression of double-stranded RNAs from the WCR gene DvSnf7, a component of the ESCRT-III complex (endosomal sorting complex required for transport), shows promising results [29, 30]. It has been associated with some commercial Cry pesticidal proteins to generate new pyramid maize hybrids (MON 87411 expressing Cry3Bb1 + DvSnf7 and SmartStax PRO expressing Cry3Bb1, Gpp34Ab1/Tpp35Ab1 and DvSnf7) [31, 32]. In addition, biological control approaches have found their niche, such as the use of entomopathogenic nematodes against WCR larvae in Europe [33]. Microbial biopesticides have also been intensively studied for biocontrol purposes, but none yet commercialized. The exceptions are microbial plant biofertilizers/biostimulants, such as different *Pseudomonas* or *Bacillus* species that may also have some pesticidal "side" effects. Additionally, non-*Bt* protein activites from *Chromobacterium*, *Brevibacillus* and *Pseudomonas* species [34–37] have been shown to successfully reduce WCR root damage in GM maize in the field, but so far there are no commercial products. Plant-derived compounds have also been evaluated for their potential to control insects [38, 39].

Repellent effects of some compounds have been demonstrated on WCR larvae in the lab [40, 41] and more recently under field conditions [42].

In the perspective of discovering novel pesticidal proteins or biopesticidal bacteria against WCR, we sequenced thousands of bacteria genomes from different origins. Then candidate pesticidal proteins were searched for based on their sequence similarities with known proteins linked to insect resistance. Subsequently, artificial diet-based bioassays were performed to evaluate their pesticidal activity. Here, we report the discovery of a family of pesticidal proteins (GDI0005A & GDI0006A) present in several *Chryseobacterium* species and having an activity against WCR larvae. Insect bioassays were performed with different WCR colonies, some resistant to commercialized pesticidal toxins, in order to judge the potential uniqueness of the MoA of the novel proteins. Also, non-targets, such as lepidopterans were tested to better understand the specificity of the discovered proteins. These findings are hoped to help pave the way for the development of novel pest management approaches urgently needed against rootworms.

## Results

### Discovery of novel pesticidal proteins

In the context of the search for pesticidal proteins active against WCR, we sequenced the genomes of several thousand bacterial strains collected in soils and from the CEA-LEMIRE collection. The predicted proteins from these genomes were compared by sequence similarity to pesticidal proteins extracted from several databases: BPPRC, the UniProtKB [43] and public patent databases such as from the European Patent office or the United States Patent and Trademark Office. Proteins with a homology level of >30% identity with known pesticidal proteins were selected for activity assessment by testing *E. coli* lysates expressing the candidate proteins against WCR. Using this strategy, two pesticidal candidates, referred as GDI0005A and GDI0006A, were identified in a *Chryseobacterium arthrosphaerae* strain (ZSTG2175), isolated from the Sahara Desert. The 16S rRNA sequences from this strain had 99% identity to those of *C. arthrosphaerae* CC-VM-7 (*NCBI:txid651561;* [44], isolated from the faeces of the pill millipede *Arthrosphaera magna* Attems [45]). The two proteins are found in the same operon. They have a low identity of less than 40% to the pesticidal proteins of the PIP45 and PIP74 families, described in [46, 47], and hardly any identity to other known pesticidal protein families. The two candidates GDI0005A and GDI0006A were named by the BPPRC Xpp84Aa1and and Xpp85Aa1 respectively due to their uncharacterized structure.

In detail, GDI0005A shares 33%, 36%, and 34% identity with PIP-45Aa-1, PIP-45Ga-1 and PIP-74Aa1, respectively (Table 1), whereas GDI0006A shares 27%, 28% and 28% identity with PIP-45Aa-2, PIP-45-Ga-2 and PIP-74Aa-2, respectively (Table 2). Despite the low level of identity, several amino acids are conserved along the sequence alignment suggesting that all members could belong to a similar structural family of proteins (S1 and S2 Figs). All these members of the PIP-45 and PIP-74 protein families were reported as active when tested in pairs. Each of the two components were encoded by genes found in the same operon of the original bacteria (S3 Fig). Some pairs were claimed to be active against WCR larvae, for instance, PIP-74Aa-1 and PIP-74Aa-2 from *Pseudomonas rhodesiae* or PIP-45Aa1 and PIP-45Aa2 from *Pseudomonas brenneri* [46]. Conversely, despite belonging to the same family of proteins, no WCR activity was reported for PIP-45-Ga1 and PIP-45-Ga-2 isolated from *Cellvibrio japonicus* [46].

GDI0005A and GDI0006A genes are organized in an operon structure with a 4 bp overlap, like PIP45 members, which strengthened the hypothesis of a binary protein with insecticidal activity (Fig 1A). The GDI0005A and GDI0006A genes are 1743 and 1404 nucleotides in

**Table 1. Percentage of amino acid identity between members of the PIP45, PIP75 and the novel discovered GDI0005A protein families.**

| | GDI0005A | GDI0103A | GDI0175A | GDI0177A | GDI0179A | GDI0181A | GDI0183A | GDI0185A | GDI0187A | PIP-45Aa-1 | PIP-45Ad-1 | PIP-45Ba-1 | PIP-45Bd-1 | PIP-45Be-1 | PIP-45Ca-1 | PIP-45Cb-1 | PIP-45Ea-1 | PIP-45Ga-1 | PIP-74Aa-1 |
|---|---|---|---|---|---|---|---|---|---|---|---|---|---|---|---|---|---|---|---|
| GDI0005A | 100 | 32 | 76 | 76 | 76 | 72 | 74 | 81 | 89 | 33 | 32 | 33 | 32 | 33 | 33 | 32 | 31 | 36 | 34 |
| GDI0103A | | 100 | 32 | 33 | 33 | 34 | 33 | 33 | 33 | 36 | 36 | 34 | 34 | 34 | 34 | 34 | 33 | 30 | 34 |
| GDI0175A | | | 100 | 93 | 86 | 84 | 84 | 76 | 76 | 32 | 31 | 32 | 32 | 33 | 33 | 32 | 29 | 36 | 33 |
| GDI0177A | | | | 100 | 87 | 84 | 84 | 76 | 77 | 32 | 32 | 33 | 33 | 33 | 33 | 33 | 29 | 36 | 33 |
| GDI0179A | | | | | 100 | 84 | 84 | 76 | 76 | 33 | 33 | 34 | 34 | 34 | 33 | 33 | 31 | 36 | 34 |
| GDI0181A | | | | | | 100 | 82 | 74 | 74 | 32 | 31 | 32 | 32 | 32 | 33 | 32 | 30 | 36 | 33 |
| GDI0183A | | | | | | | 100 | 73 | 75 | 32 | 32 | 33 | 33 | 33 | 33 | 33 | 31 | 36 | 33 |
| GDI0185A | | | | | | | | 100 | 82 | 32 | 32 | 33 | 32 | 33 | 33 | 32 | 30 | 36 | 34 |
| GDI0187A | | | | | | | | | 100 | 33 | 33 | 33 | 33 | 33 | 33 | 33 | 32 | 37 | 33 |
| PIP-45Aa-1 | | | | | | | | | | 100 | 97 | 89 | 88 | 87 | 78 | 77 | 60 | 33 | 31 |
| PIP-45Ad-1 | | | | | | | | | | | 100 | 87 | 88 | 86 | 77 | 76 | 59 | 32 | 30 |
| PIP-45Ba-1 | | | | | | | | | | | | 100 | 96 | 84 | 77 | 77 | 60 | 33 | 30 |
| PIP-45Bd-1 | | | | | | | | | | | | | 100 | 84 | 77 | 76 | 59 | 33 | 30 |
| PIP-45Be-1 | | | | | | | | | | | | | | 100 | 75 | 73 | 58 | 32 | 31 |
| PIP-45Ca-1 | | | | | | | | | | | | | | | 100 | 95 | 61 | 33 | 31 |
| PIP-45Cb-1 | | | | | | | | | | | | | | | | 100 | 59 | 32 | 30 |
| PIP-45Ea-1 | | | | | | | | | | | | | | | | | 100 | 31 | 30 |
| PIP-45Ga-1 | | | | | | | | | | | | | | | | | | 100 | 35 |
| PIP-74Aa-1 | | | | | | | | | | | | | | | | | | | 100 |

Multiple Alignment with free end gaps were produced using Blosum62 matrix on Geneious Prime software. Information on the origin of protein sequences is listed in the S1 Table.

length (including a stop codon) resulting in full-length proteins of 580 and 467 amino acids with a theoretical molecular weight of 64.4 kDa and 52.3 kDa, respectively. A scan for matches against the InterPro database [47] did not detect any known domain or motif for GDI0005A. In contrast, GDI0006A contains a non-cytoplasmic domain predicted by Phobius-1.01 [48] that covers the entire protein apart from an N-terminal signal peptide (SP) found in prokaryotic lipoproteins (Fig 1B). Based on the SignalP-5.0 software [49], the SP is predicted to have a size of 18 amino acids (AA) with a cleavage site between serine 15 and cysteine 16. The first AA of the mature protein is therefore predicted to be a cysteine, which is a lipoprotein feature [50]. The rest of the protein is predicted as a non-cytoplasmic domain suggesting that the protein might be a membrane-associated protein. These results strengthen the hypothesis of a lipoprotein anchored in the outer membrane of the original bacteria that could explain the lower yield of GDI0006A compared to GDI0005A when produced in *Escherichia coli*. No matches with any known 3-D protein structure for either protein were found using the Phyre2 web portal for protein modeling, prediction and analysis [51].

## GDI0005A and GDI0006A are binary pesticidal proteins having WCR activity

Finding a pair of proteins with low similarity with PIP-45 / PIP-74 proteins and encoded by genes in the same operon is not sufficient to predict that they will be active against WCR. The potential WCR activity of GDI0005A in association with GDI0006A was therefore tested to

**Table 2. Percentage of amino acid identity between members of the PIP45, PIP75 and novel discovered GDI0006A protein families.**

| | GDI0006A | GDI0104A | GDI0176A | GDI0178A | GDI0180A | GDI0182A | GDI0184A | GDI0186A | GDI0188A | PIP-45Aa-2 | PIP-45Ad-2 | PIP-45Ba-2 | PIP-45Bd-2 | PIP-45Be-2 | PIP-45Ca-2 | PIP-45Cb-2 | PIP-45Ea-2 | PIP-45Ga-2 | PIP-74Aa-2 |
|---|---|---|---|---|---|---|---|---|---|---|---|---|---|---|---|---|---|---|---|
| **GDI0006A** | 100 | 29 | 70 | 69 | 68 | 67 | 68 | 77 | 90 | 27 | 27 | 28 | 28 | 27 | 29 | 29 | 30 | 28 | 28 |
| **GDI0104A** | | 100 | 29 | 29 | 28 | 29 | 29 | 29 | 29 | 28 | 28 | 28 | 29 | 29 | 28 | 28 | 31 | 28 | 27 |
| **GDI0176A** | | | 100 | 98 | 87 | 85 | 89 | 70 | 70 | 27 | 27 | 28 | 29 | 28 | 29 | 29 | 29 | 27 | 30 |
| **GDI0178A** | | | | 100 | 87 | 84 | 89 | 70 | 70 | 28 | 28 | 28 | 28 | 28 | 29 | 29 | 29 | 27 | 30 |
| **GDI0180A** | | | | | 100 | 84 | 87 | 70 | 69 | 26 | 27 | 27 | 28 | 27 | 28 | 28 | 28 | 28 | 28 |
| **GDI0182A** | | | | | | 100 | 84 | 69 | 67 | 27 | 27 | 27 | 28 | 27 | 28 | 28 | 29 | 27 | 29 |
| **GDI0184A** | | | | | | | 100 | 69 | 69 | 27 | 27 | 28 | 28 | 28 | 29 | 29 | 29 | 28 | 28 |
| **GDI0186A** | | | | | | | | 100 | 78 | 27 | 27 | 28 | 28 | 27 | 29 | 29 | 29 | 28 | 28 |
| **GDI0188A** | | | | | | | | | 100 | 27 | 27 | 27 | 28 | 28 | 28 | 29 | 29 | 28 | 28 |
| **PIP-45Aa-2** | | | | | | | | | | 100 | 94 | 84 | 84 | 77 | 66 | 66 | 46 | 27 | 25 |
| **PIP-45Ad-2** | | | | | | | | | | | 100 | 84 | 83 | 78 | 67 | 67 | 46 | 27 | 25 |
| **PIP-45Ba-2** | | | | | | | | | | | | 100 | 96 | 75 | 66 | 66 | 47 | 28 | 25 |
| **PIP-45Bd-2** | | | | | | | | | | | | | 100 | 75 | 66 | 67 | 47 | 29 | 25 |
| **PIP-45Be-2** | | | | | | | | | | | | | | 100 | 68 | 67 | 46 | 28 | 25 |
| **PIP-45Ca-2** | | | | | | | | | | | | | | | 100 | 93 | 45 | 29 | 26 |
| **PIP-45Cb-2** | | | | | | | | | | | | | | | | 100 | 44 | 29 | 25 |
| **PIP-45Ea-2** | | | | | | | | | | | | | | | | | 100 | 28 | 25 |
| **PIP-45Ga-2** | | | | | | | | | | | | | | | | | | 100 | 24 |
| **PIP-74Aa-2** | | | | | | | | | | | | | | | | | | | 100 |

Multiple Alignment with free end gaps were produced using Blosum62 matrix on Geneious Prime software. Information on the origin of protein sequences is listed in the S1 Table.

confirm the information obtained from sequence analysis. GDI0005A and GDI0006A were produced as His-tagged recombinant proteins in *E. coli* and *Pseudomonas fluorescens* expression systems. Recombinant proteins were detected from crude lysates by western blotting with His-antibodies. Bioassays with artificial diet overlaid with crude lysates from *E. coli* and *P. fluorescens* containing GDI0005A and GDI0006A revealed significant mortality of WCR neonates and most surviving larvae were stunted (Fig 2). In contrast, individual crude lysates containing either GDI0005A or GDI0006A alone did not show any activity (Fig 2) demonstrating that both GDI0005A and GDI0006A are required for full activity against WCR larvae.

The binary pesticidal proteins GDI0005A and GDI0006A were also tested against the major lepidopteran maize pests: European corn borer (ECB) *Ostrinia nubilalis* (Hubner), fall armyworm (FAW) *Spodoptera frugiperda* (J.E. Smith), southwestern corn borer (SWCB) *Diatraea grandiosella* Dyar, and corn earworm (CEW) *Helicoverpa zea* (Boddie), using diet overlay assays. No significant activity was detected against any of these lepidopterans (S2 Table).

## GDI0005A and GDI0006A have a putative novel mode of action (MoA)

To determine whether our binary pesticidal proteins are novel to the current commercial proteins Gpp34Ab1/Tpp35Ab1 and Cry3Bb1, the crude lysates of recombinant GDI0005A and GDI0006A proteins from *E. coli* were tested in bioassays against WCR colonies with field-evolved resistance to Gpp34Ab1/Tpp35Ab1 and Cry3Bb1. The Gpp34Ab1/Tpp35Ab1- resistant population of WCR used has an incomplete resistance to Gpp34Ab1/Tpp35Ab1 toxins

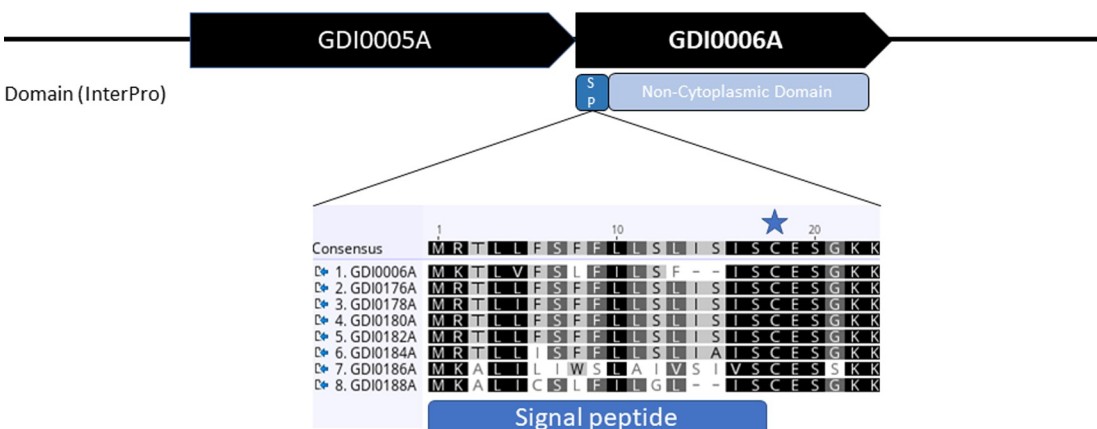

**Fig 1. Characteristics of the GDI0005A and GDI0006A operon from *Chryseobacterium arthrosphaerae* ZSTG2175 genome.** **(A)** The two GDI0005A and GDI0006A open reading frames overlap by 4 nucleotides. **(B)** Sequence overview of the signal peptide from all GDI00006A protein family members. The blue star indicates the potential cleavage site between the serine and the cysteine amino acid predicted by the SignalP5.0 algorithm.

[52]. As expected in bioassays, the lysate of Gpp34Ab1/Tpp35Ab1 recombinant proteins caused only a partial loss of weight (0.3 mg versus 0.5 mg) in larvae from Gpp34Ab1/

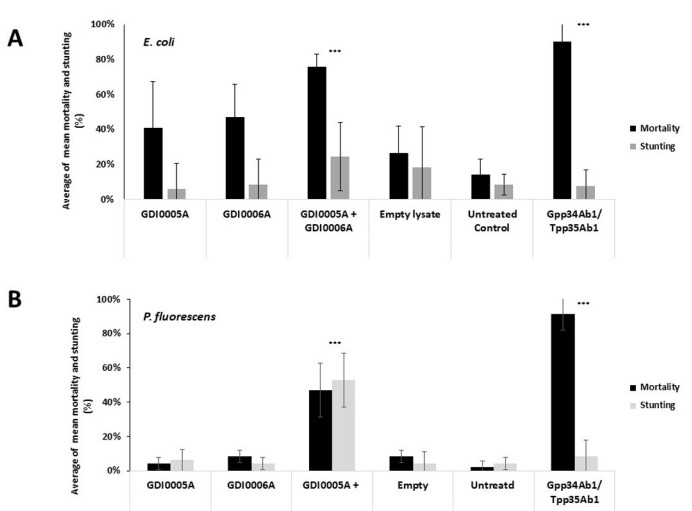

**Fig 2. Activity of GDI0005A and GDI0006A proteins against western corn rootworm (WCR).** In *Escherichia coli* **(A)** and *Pseudomonas fluorescens* **(B)** strain lysates tested on WCR neonates (USDA ARS non-diapause susceptible colony) 5 days after treatment and infestation using artificial diet overlay bioassays. Activity was evaluated by assessing the percentage of mortality (black boxes) and the percentage of stunting (grey boxes). Each larva was fed on ~200 μL of WCR artificial diet overlaid with 20 μL of bacterial lysate. Bacterial lysates contained GDI0005A, GDI0006A or both proteins mixed. Negative controls consisted of lysates from bacteria transfected with the empty expression plasmid (empty lysate) and sterilized water (untreated control). The positive control consisted of lysates from bacteria transfected with a plasmid expressing the commercial Gpp34Ab1/Tpp35Ab1 binary proteins. Each experiment consisted of at least six 96-well plates each with 8 wells per treatment, totaling a sample size of 48 per treatment per experiment. A Fisher exact test was used to compare treatment results to the empty lysate at significance levels of $P < 0.05^*$ and $<0.005^{**}$. The untreated control was only used to evaluate the quality and reliability of each bioassay. The error bars show the standard error of the mean (SEM).

Tpp35Ab1 resistant colonies. On the contrary, they caused a nearly complete weight loss (<0.02 mg versus 0.5 mg) for Cry3Bb1 resistant colonies (Fig 3). When the lysate of GDI0005A and GDI0006A recombinant proteins was tested, a high weight loss (<0.06 mg versus 0.5 mg) was observed on both Gpp34Ab1/Tpp35Ab1 and Cry3Bb1 resistant strains (S4 Fig). These results suggest that there is no cross-resistance between GDI0005A/GDI0006A and the two commercial *Bt*—pesticidal proteins against WCR. Thus, GDI0005A and GDI0006A may not share the same binding sites as Gpp34Ab1/Tpp35Ab1 or Cry3Bb1 which could imply a new MOA.

## GDI0005A/GDI0006A homologs from *Chryseobacterium* species have WCR activity

To investigate the occurrence of the novel pesticidal proteins GDI0005A/GDI0006A, a BlastP search against public (Uniprot) and proprietary databases was conducted leading to the identification of seven homologs to GDI0005A ranging from 72% identity (GDI0181A) to 89% identity (GDI0187A) and to GDI0006A ranging from 67% identity (GDI0182A) to 90% identity (GDI0188A), respectively (Table 1). These homologs were all found in *Chryseobacterium* species isolated from highly diverse environments such as soil, plant rhizosphere, raw chicken, and a lactic acid beverage (Material & Methods, Table 3). Figs 4 and 5 show the protein sequence alignments of GDI0005A and GDI0006A with their homologs and the corresponding phylogenetic trees. The differences are mainly located in the N- and C-terminal parts of the protein, the GDI0185A/GDI0186A proteins being the most divergent. The SP and the non-cytoplasmic domain detected in GDI0006A previously were also detected in all

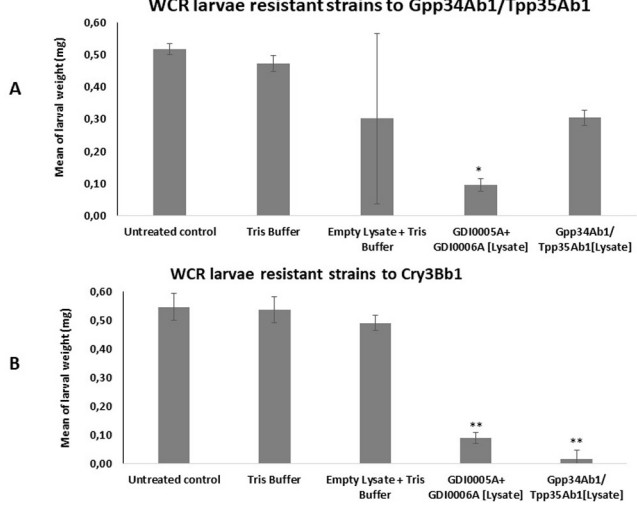

**Fig 3. Activity of GDI0005A and GDI0006 proteins produced in *Escherichia coli* on resistant western corn rootworm (WCR) larvae using diet overlay bioassays. (A)** WCR strain resistant to the insecticidal proteins Gpp34Ab1/Tpp35Ab1. **(B)** WCR strain resistant to the insecticidal protein Cry3Bb1. Each larva was exposed to ~200 μL of WCR artificial diet overlaid with 20 μL of bacterial lysate containing (GDI0005A + GDI0006A). Negative controls consisted of diet overlaid with lysates from *E. coli* bacteria transfected with the empty expression plasmid (empty lysate) or Tris buffer and sterilized water (untreated control). The positive control consisted of diet overlaid with lysates from bacteria transfected with a plasmid expressing the commercial Gpp34Ab1/Tpp35Ab1 binary proteins. Each experiment consisted of at least three 96-well plates each with 8 wells per treatment, totaling a sample size of 24 per treatment per experiment. Mortality and larval weight was assessed 5 days after treatment. One-way analysis of variance (ANOVA) and Tukey's HSD post-hoc test were used to assess effects of treatments and differences among the treatments at a significance level of P < 0.01 **. The error bars show the standard deviation.

**Table 3. Origin and characteristics of the *Chryseobacterium* strains studied for pesticidal activity.**

| Strain name | Available strains (ID Collection) | Binary Toxin 1 | Binary Toxin 2 | Strain Storage | Sources |
|---|---|---|---|---|---|
| *Chryseobacterium arthrosphaerae* | ZSTG2175 | **GDI0005A** | **GDI0006A** | LEMIRE | Isolated from the Sahara Desert (W. Achouak, Lemire, CEA Cadarache) |
| *Chryseobacterium arthrosphaerae CC-VM* | DSM 25208[T] | **GDI0005A1** | **GDI0006A1** | PROTEUS | Jeong, J.-J., Lee, D.W., Park, B., Sang, M.K., Choi, I.-G., and Kim. K.D. "Chryseobacterium cucumeris sp. nov., an endophyte isolated from cucumber (Cucumis sativus L.) root and emended description of Chryseobacterium arthrosphaerae." Int. J. Syst. Evol. Microbiol. (2017) 67:610–616. |
| *Chryseobacterium carnipullorum* | DSM 25581[T] | **GDI0175A** | **GDI0176A** | ¨PROTEUS | Raw chicken from a poultry processing plant in Bloemfontein in South Africa (Charimba et al., 2013) |
| Chryseobacterium *shigense* | DSM 17126[T] | **GDI0177A** | **GDI0178A** | ¨PROTEUS | Lactic acid beverage (Shimomura *et al.*, 2005 [86]) |
| *Chryseobacterium sp. OV279 (CFB group bacteria)* | Not available | **GDI0179A** | **GDI0180A** | NA | Populus root rhizosphere NCBI Uniprot |
| *Chryseobacterium sp. YR203* | Not available | **GDI0181A** | **GDI0182A** | NA | Populus root rhizosphere NCBI Uniprot |
| *Chryseobacterium kwangjuense* | JCM 15904[T] | **GDI0183A** | **GDI0184A** | ¨PROTEUS | Root of a pepper plant Capsicum annuum in Kwangju in Korea (Sang et al., 2013 [85]) |
| *Chryseobacterium sp. OV705* | Not available | **GDI0185A** | **GDI0186A** | NA | Populus root rhizosphere *NCBI Uniprot* |

GDI0006A-like sequences. A C-terminal extension (Cter) of 10 amino acids was only detected in the GDI0006A sequence.

All homologs of GDI0005A and GDI0006A were expressed as recombinant proteins in *E. coli* and tested for WCR activity. Results showed that all homologs when tested as protein pairs had a similar pesticidal activity (mortality and stunting effects) as GDI0005A and GDI0006A (Fig 6).

## Cross-activity between GDI0005A and GDI0006A components from different origins

PIP-45 pesticide proteins are binary toxins having an activity when a member of the PIP-45-1 family is associated with the corresponding member of the PIP-45-2 family (*i.e.* found in the same strain). However, members of the PIP-45-1 family have been shown to be active on WCR when associated with different members of the PIP-45-2 family [46] which are not from the same strains. We tested whether a similar situation would apply to the GDI0005A / GDI0006A binary proteins. We set up a series of cross-experiments between GDI0005A and different GDI0006A-like proteins following the diet assay methodology as described above. Whereas two combinations (GDI0005A/GDI0180A and GDI0005A/GDI0182A) did not show

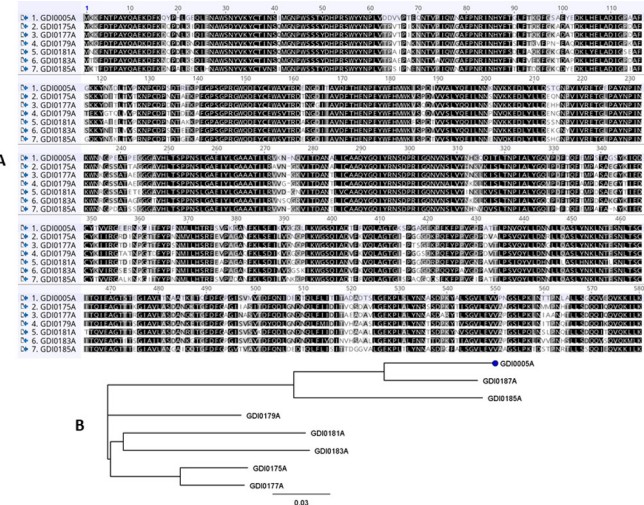

**Fig 4. Protein alignment of GDI0005A homologs. (A)** GDI0005A and homologous protein sequences were aligned using Geneious Multiple alignment tool with the Blosum 62 cost matrix. Identical amino acids are indicated in black; conserved amino acids in grey; all the others are in white. **(B)** Phylogenetic tree of GDI0005A homologs built with the Neighbor-joining method using the Jukes-Cantor Genetic Distance model (Geneious Tree Builder from Geneious).

any activity, all other combinations, GDI0005A/GDI0176A, GDI0005A/GDI0178A, GDI0005A/GDI0184A and GDI0005A/GDI0186A provided WCR activity (Fig 7). Therefore, as with PIP-45, different sources of GDI0006A from *Chryseobacterium* species can provide pesticidal activity with GDI0005A.

## The N-terminal sequence is essential for the activity of GDI0006A

The N-terminal signal peptide of 18 AA present all GDI0006A homologs and the C-terminal extension of 10 AA present only in GDI0006A were evaluated for their role in the activity of

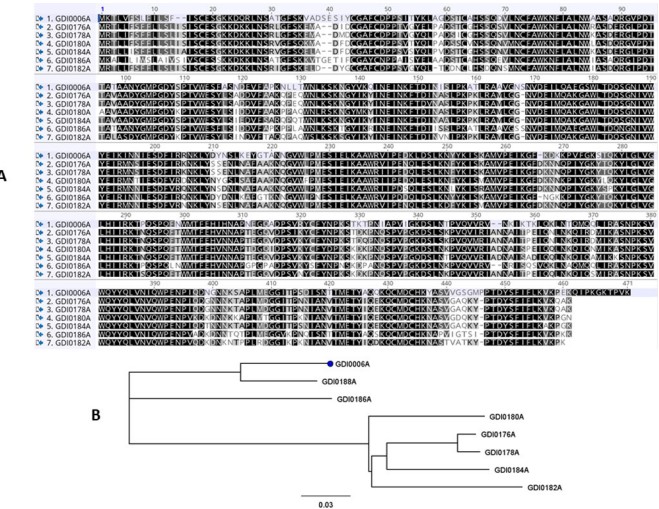

**Fig 5. Protein alignment of GDI0006A homologs. (A)** GDI0006A and homologous protein sequences were aligned using Geneious Multiple alignment tool with the Blosum 62 cost matrix. Identical amino acids are indicated in black; conserved amino acids in grey; all the others are in white. **(B)** Phylogenetic tree of GDI0006A homologs built with the Neighbor-joining method using the Jukes-Cantor Genetic Distance model (Geneious Tree Builder from Geneious).

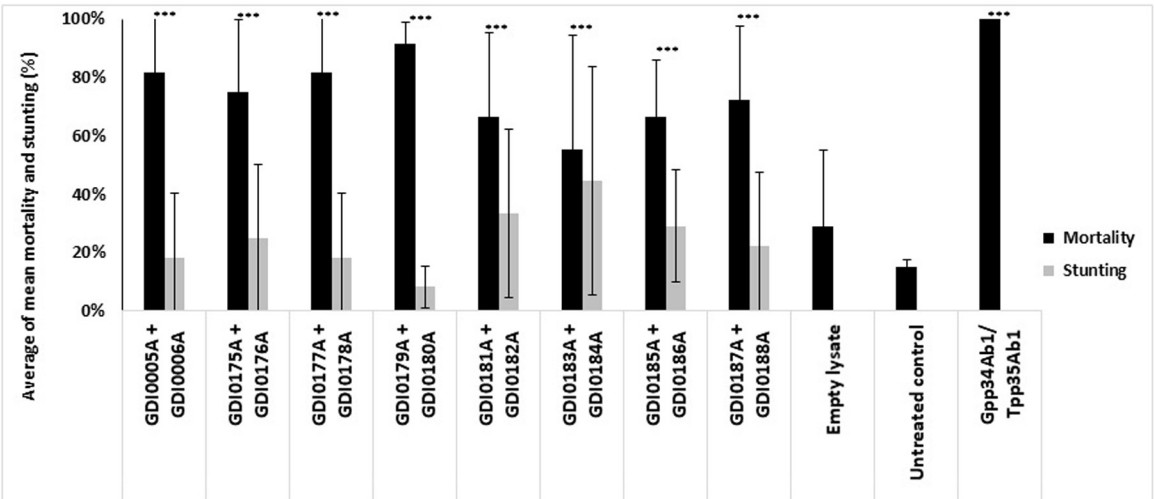

**Fig 6. Activity of different pairs of GDI0005A/GDI0006A proteins, produced in *E. coli* on WCR susceptible neonates 5 days after treatment and infestation using a diet overlay bioassay.** Different GDI0005A and GDI0006A homologs are compared. Activity was evaluated by assessing the percentage of mortality (black boxes) and the percentage of stunting (grey boxes). Each WCR larvae was exposed to ~200 µL of WCR artificial diet overlaid with 20 µL of bacterial lysate mixtures containing (GDI0005A + GDI0006A) or (GDI0005A homologs + GDI0006A homologs). Negative controls were lysates from *E. coli* bacteria transfected with the empty expression plasmid (empty lysate) and sterilized water (untreated control). The positive control was a lysate from bacteria transfected with a plasmid expressing the commercial Gpp34Ab1/Tpp35Ab1 binary proteins. Each experiment consisted of at least six 96-well plates each with 8 wells per treatment, totaling a sample size of 48 per treatment per experiment. A Fisher exact test was used to compare treatment results to the empty lysate at significance levels of P < 0.05*, < 0.005**, <0.001***. The untreated control was only used to evaluate the quality and reliability of each bioassay. The error bars show the standard error of the mean (SEM).

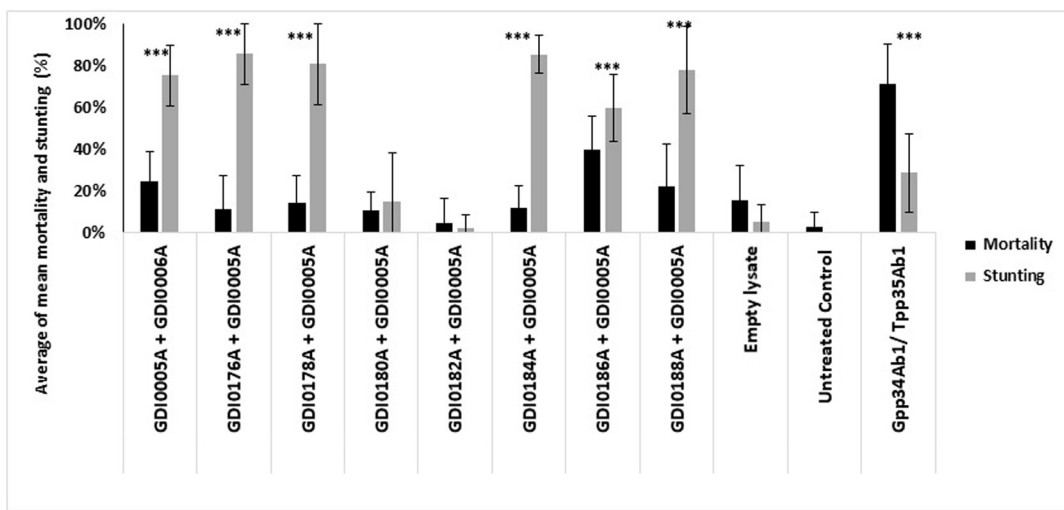

**Fig 7. Cross activity between component I (GDI0005A and homologues) and component II (GDI0006A and homologues).** Activity was evaluated on neonates of western corn rootworm (WCR) 5 days after treatment and infestation by assessing the percentage of mortality (black boxes) and the percentage of stunting (grey boxes). Each larva was exposed to ~200 µL of WCR artificial diet overlaid with 20 µL of bacterial lysate mixtures containing (GDI0005A+GDI0006A) or (GDI0005A combined with GDI0006A homologs). Negative controls were lysates from *E. coli* bacteria transfected with the empty expression plasmid (empty lysate) and sterilized water (untreated control). The positive control were lysates from bacteria transfected with a plasmid expressing the commercial Gpp34Ab1/Tpp35Ab1 binary proteins. Six 96-well plates each with 8 wells per treatment were used, totaling a sample size of 48 per treatment. A Fisher exact test was used to compare treatment results to the empty lysate at significance levels of P < 0.05*, < 0.005**, <0.001***. The untreated control was only used to evaluate the quality and reliability of each bioassay. The error bars show the standard error of the mean (SEM).

GDI0006A. Six variants (named V1 to V6) of GDI0006A with or without the SP and C-terminal sequences were generated (Fig 8A). All constructs contained a His-tag fused to their N- or C-terminus for detection of the recombinant protein by western-blot using an anti-histidine antibody. GDI0006A was present in all bacterial lysates but in variable amounts depending on the construct (Fig 8B). The lack of signal in the sample containing the complete GDI0006A produced with a N-term His-tag (row 6, Fig 8B) reinforces the hypothesis that the native GDI0006A is normally processed through the cleavage of its SP (deletion of the His-tag during the cleavage process).

All the bacterial lysates were then tested associated with GDI0005A on WCR larvae using diet overlay bioassays. The lysates obtained from the V1 to V4 constructs where SP was deleted were all inactive, whereas those from V5 and V6 harboring SP showed activities (Fig 8C). Finding an activity for both V5 and V6 lysates indicates that the position of the His-tag had no effect on the protein activity. These results demonstrate that the N-terminal SP, presumably required for targeting to the membrane, appears to be essential for the activity of the GDI0006A protein. On the contrary, the C-terminal sequence has no impact on the activity of the protein, as shown by the results with GDI0006 homologs that do not have a C-terminal sequence extension.

## Cultured *Chryseobacterium. arthrosphaerae* strains have WCR activity

The original *C. arthrosphaerae* strain isolated from the Sahara Desert (ZSTG2175) and the strain isolated from pill millipede (DSM25208) were tested for their activity against WCR. A PCR assay from liquid cultures confirmed the presence of the genes encoding GDI0005A and

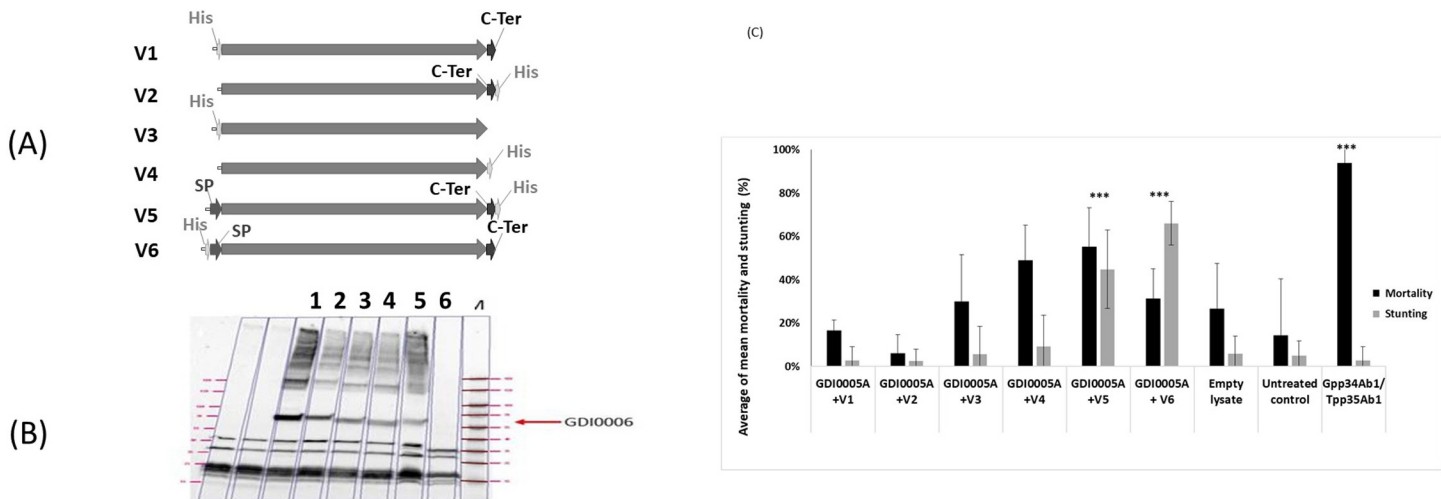

**Fig 8. Expression and activity of GDI0006A. (A)** Representation of GDI0006A new constructs. V1: signal peptide (SP) deleted, His-tag at the N-terminus; V2: signal peptide deleted and His-tag at C-terminus; V3: signal peptide deleted, C-terminal extension deleted, and His-tag at N-terminus; V4: signal peptide deleted, C-terminal extension deleted and His-tag at C-terminus; V5: full protein with a His-tag at the C-terminus; V6: full protein with a His-tag at the N-terminus (as in the original construct used in bioassays). **(B)** Western blot analysis of GDI0006A construct expression using an anti-His antibody. Lane 1 corresponds to the construct V1, lane 2 to construct V2, lane 3 to construct V3, lane 4 to construct V4, lane 5 to construct V5 and lane 6 to construct V6. **(C)** Activity of GDI0006A lysates from constructs (V1 to V6) mixed with GDI0005A lysates on WCR neonates 5 days after treatment and infestation using diet overlay bioassay. Activity was evaluated by assessing the percentage of mortality (black boxes) and the percentage of stunting (grey boxes). Each WCR larvae was exposed to ~200 μL of WCR artificial diet overlaid with 20 μL of bacterial lysate mixtures containing (GDI0005A + GDI0006A different constructs). Negative controls were lysates from *E. coli* bacteria transfected with the empty expression plasmid (empty lysate) and sterilized water (untreated control). The positive control was a lysate from bacteria transfected with a plasmid expressing the commercial Gpp34Ab1/Tpp35Ab1 binary proteins. Six 96-well plates each with 8 wells per treatment were used, totaling a sample size of 48 per treatment. A Fisher exact test was used to compare treatment results to the empty lysate at significance levels of P < 0.05*, < 0.005**, <0.001***. The untreated control was only used to evaluate the quality and reliability of each bioassay.

GDI0006A in both strains and sequencing of the PCR products confirmed that both strains carry identical sequences (S3 Table).

Two independent experiments were conducted with each strain grown in rich medium. The supernatant and lysates from the pellets of both cultures were tested for their activity on WCR larvae. Experiments revealed that most samples from ZSTG2175 or DSM25208 bacteria caused mortality and stunted growth in WCR neonates within 5 days (Fig 9) except for two samples from DSM25208. However, the ZSTG2175 strain overall showed statistically significantly more activity than the DSM25208 strain with both supernatant and pellet samples.

## Discussion

Insect pressure in areas of the world where maize is grown extensively is usually high [53]. Depending on the climatic conditions, insect pests can have a major impact on yield and food security [2]. WCR is one of the most damaging insects of the maize crop in North America and Europe. In addition to synthetic insecticides and biocontrol agents such as entomopathogenic nematodes, genetically engineered maize hybrids are currently available to manage WCR populations in countries such as the USA. The latter express Cry3 proteins (including Cry3Bb, mCry3A and eCry3.1Ab) and Gpp34Ab1/Tpp35Ab1 from *B. thuringiensis*. However, the ability of WCR to develop resistance to these commercially-deployed pesticidal proteins is impacting the efficiency of these products [54]. Thus, finding candidates with new MoAs is crucial to develop new effective products. Whereas many studies still focus on *B. thuringiensis* to find new pesticidal proteins, several novel pesticidal proteins from non-*Bt* strains have been recently identified. These include IPD072Aa, a small protein of 86 amino acids purified from *Pseudomonas chlororaphis* [36], Mpf3Aa1 (formerly GNIP1a) from *Chromobacterium piscinae* having a Membrane Attack Complex/Perforin (MACPF) domain that is also present in proteins involved in animal immune response [35], PIP-47Aa from *Pseudomonas mosselii*, which

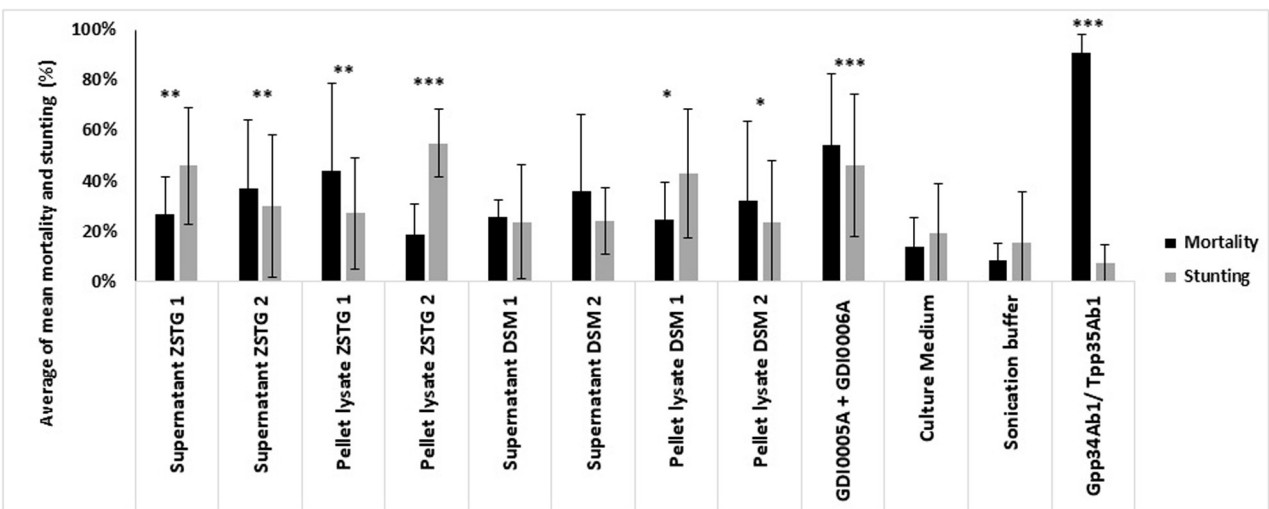

**Fig 9. Assessment of the activity of 2 strains of *Chryseobacterium arthrosphaerae* (ZSTG2175 and DSM25208) on WCR neonates.** Activity was evaluated 5 days after treatment and infestation using a diet overlay bioassay by assessing the percentage of mortality (black boxes) and the percentage of stunting (grey boxes). Each WCR larvae was exposed to ~200 µL of WCR artificial diet overlaid with 20 µL of supernatant or pellet lysates of both strains (ZSTG 1 and DSM 1: first replicate of ZSTG2175 and DSM25208; ZSTG 2 and DSM 2: second replicate of ZSTG2175 and DSM25208). Negative controls were the culture medium and sonication buffer. The positive control were lysates from *E. coli* transfected with a plasmid expressing the commercial Gpp34Ab1/Tpp35Ab1 binary proteins. For each experiment, a total of 48 WCR neonate larvae were analyzed. A Fisher exact test was used to compare treatment results to the culture medium at significance levels of $P < 0.05^*$, $< 0.005^{**}$, $<0.001$. The error bars show the standard error of the mean (SEM).

does not share domains with other known proteins [37] and Mpp75Aa protein homologs from *Brevibacillus laterosporus* belonging to the Etx/MTX2 family [34]. All these 4 proteins were reported to kill WCR larvae in laboratory assays and seem specific to a few coleopteran species. In all these studies, transgenic maize plants expressing these novel pesticidal proteins showed protection from root damage by WCR demonstrating that proteins from other bacterial species could be as efficacious as *Bt* pesticidal proteins in controlling major insect pests such as WCR.

In this study, we report the discovery of an entirely new family of binary pesticidal proteins isolated from several *Chryseobacterium* species. We demonstrated that the two newly discovered GDI0005A and GDI0006A proteins, such as from *C. arthrosphaerae*, exhibit a WCR activity when both proteins were ingested by the pest larvae. These proteins belong to a larger protein group that was first identified by Gruver et al. [46] as a two-component system with pesticidal activity (S1 Fig). These proteins were identified in many Proteobacteria such as *Pseudomonas* species *(P. putida*, *P. monteilii*, *P. brenneri*, *P. fluorescens)* from the soil or water, but also in other genera like *Burkholderia, Cellvibrio, Paracoccus* in addition to *Chryseobacterium* (this study). The pesticidal activity is found in several clades in the dendrogram, although the level of similarity between clades is low. According to our study and in addition to the results of Gruver et al. [46], no activity against lepidopteran maize pests such as CEW, FAW, ECB, SWCB was detected. Thus, the pesticidal activity of GDI0005A/GDI0006A could be limited to certain coleopteran species, but further testing is needed to define the spectrum of activity more precisely.

Although we have identified GDI0005A and GDI0006A homologs in several *Chryseobacterium* species, not all members of this genus possess these genes indicating that they are probably part of the nonessential genome of *Chryseobacterium*. This is typical of genes encoding nonessential function for the survival of the microorganism like pesticidal proteins (absence of pesticidal encoding genes in the list of essential genes in DEG database/ [55]). However, they may be involved in the adaptation of the bacterium to its ecological niche and to its fitness.

To commercialize maize plants expressing pesticidal proteins with an activity against a maize pest, a prerequisite is to determine whether the MoA of the pesticidal proteins is different to the MoA of an existing commercial product. Apart from the RNAi mode of action, only two different MoAs of microbial pesticidal proteins are present in commercial WCR products: one from the binary pesticidal protein Gpp34Ab1/Tpp35Ab1 and the other from different Cry3 variants [56]. GDI0005A/ GDI0006A recombinant proteins when overlaid on WCR diet, induced a significant weight loss in populations resistant to Gpp34Ab1/Tpp35Ab1 and Cry3Bb1 (Fig 3). Even if the level of mortality caused by GDI0005A/GDI0006A was relatively low in these experiments, these results suggests that the binary pesticidal proteins are active against resistant strains and could represent an additional component to complement the WCR control arsenal. Additional studies such as competition experiments using brush border membrane vesicles (BBMV) could confirm whether this new family of binary toxins have a different binding site on mid-gut cell membranes compared to other commercial toxins [57].

Bacteria of the genus *Chryseobacterium* [58] can be found in various environments such as soil [59], rhizosphere [60], water (either wastewater or freshwater) [61]; chicken [62], fish [63], insect gut [64, 65] and faeces of millipedes [45]. Many of those species are agriculturally important showing efficacy as plant growth-promoting rhizobacteria [66]. An example is the biological control of the plant pathogen *Phytophthora capsici* (causative agent of *Phytophthora* blight) with *Chryseobacterium* strains ISE14 and KJ9C8 [67, 68].

When it comes to wild-type bacteria, pesticidal activities against plant pests have been described in several types of non-*Bt* bacteria. For example, cell lysates and crude protein extracts of *P. mosselii* and *P. chrlororaphis* isolates showed a strong activity against WCR [36,

37] that led to the identification of the pesticidal proteins PIP47Aa and IPD072Aa. However, the activity of *Chryseobacterium* strains against plant insects pests has only been described once [69]. In their study, *Chryseobacterium* strains Rz1.2-5A associated with avocado roots showed entomopathogenic activity against laboratory-reared avocado thrips *Scirtothrips perseae* with an efficiency similar to the chemical control. No molecules or proteins associated with this activity were described. In our study, we found that several strains of *Chryseobacterium* contain in their genome proteins with an activity against WCR larvae. The use of *Chryseobacterium* strains in biological control could be envisaged based on our results showing that some cultivated *C. arthrosphaerae* strains are active against WCR larvae. Even if the link between the protein and the strain activity has not been fully established yet, we believe our results can provide significant insights to encourage further studies on this bacterium species and its proteins to control coleopteran pests.

## Materials and methods

### Origin of the bacterial strains assessed

The characteristics of the *Chryseobacterium strains* described for pesticidal activity are listed in the Table 3.

### Medium and cultivation conditions of *Chryseobacterium* strains

*Chryseobacterium* strains were grown aerobically at 30˚C using a rich medium (Overnight Express™ Instant LB Medium–Novagen). Cultures were centrifuged for 20 min at 4000 rpm. Cell pellets were resuspended in a sonication buffer (50 mM Tris, 150 mM NaCl pH 7.5) and sonicated in an ice bath for 3 min using a 500W sonicator probe (Fisherbrand FB505AEUK-220).

### DNA extraction and sequencing

Genomic DNA was extracted from overnight cultures and quantified by Picogreen measurement. The quality was checked with an agarose gel electrophoresis analysis. The genomic DNA samples were normalized before being fragmented using Adaptative Focused Acoustics technology. Illumina-compatible PCR free libraries were produced from each initial DNA sample. Each library was individually barcoded with a unique double index strategy and then quantified by qPCR measurement before being pooled. 96 libraries were pooled together and sequenced with an Illumina paired-end 2x100 bases strategy. Roughly 50 Mb were produced per pool. Read sequences from each sample were both adapted, and quality trimmed with the tool Cutadapt version 1.8.3 (https://pypi.org/project/cutadapt/). Trimmed reads were further assembled with the de novo assembler SPADES version 3.5.0 [70] and protein coding genes were predicted with tool Prodigal version 2.6.3 [71]. Predicted protein sequences were annotated based on homology to sequences in public databases. Genomes were also analyzed using the MicroScope platform [72].

### Protein analysis tools

Multiple sequence alignments and conserved domain searches were carried out using Geneious Prime 2021.1.1 (https://www.geneious.com) and appropriate tools such as InterProScan [73] and Signal 5.0 [74]. Predictions of protein 3D structure were carried out with Phyre2 software [51].

## Recombinant protein expression in *E. coli*

All GDI sequences and Cry34Ab1/Cry35Ab1 (Gpp34Ab1/Tpp35Ab1) are listed in the S1 Table. They were obtained from genome sequencing or public databases and adapted to *E. coli* codon usage and synthesized by GenScript Biotech (Leiden, Netherlands). The corresponding DNA fragments were cloned using the Electra Protocol (https://www.atum.bio/catalog/reagents/electra#__electra-overview) into the pD441-NH vector (ATUM, Newark, California, USA) under the control of the IPTG-inducible T5 promoter. Cloning was performed in *E. coli* BL21DE3 strain which is suitable for protein expression. The GDI0006A variants were obtained by PCR from the original construct, for versions with a C-terminus tag, cloning was performed in the pD441-CH vector (ATUM, Newark, California, USA). All the plasmids were verified by sequencing. Protein expression was performed overnight at 30˚C in Staby Switch auto inducible expression medium (DelphiGenetics, Charleroi, Belgium) under agitation at 200 rpm in a Infors HT Multitron Pro incubator (Infors AG, Bottmingen, Switzerland).

## Recombinant protein expression in *Pseudomonas fluorescens*

The bacterial strain and plasmid used in this study were described in [75]. *P. fluorescens* KOB2Δ1, is the *alkB* gene knockout mutant of *P. fluorescens* CHA0 and the plasmid pCom10, is the pCom8 plasmid modified to carry kanamycin resistance instead of gentamycin resistance. The recombinant cells of *P. fluorescens* KOB2Δ1 transformed with the pCOM10 plasmid were grown in Luria-Bertani (LB) medium with kanamycin (60 μg/mL) at 30˚C. Synthetic genes were codon optimized for *P. fluorescens* and cloned in pCOM10 between *NdeI* and *BamHI* sites and plasmids were checked by sequencing. The recombinant cells were transformed by electroporation (Eppendorf-electroporator 2510, 1600V) with 500 ng of each plasmid. Transformations were plated on LB agar plates containing kanamycin and glucose (1%). Expression cultures were performed in LB medium with kanamycin (60 μg/mL) at 30˚C to reach OD600 of 0.4–0.6. Then cultures were induced with 0.05% (w/v) dicyclopropyl ketone (DCPK) and incubated 20 h at 30˚C. Cells pellets were resuspended in buffer Tris-HCl 20 mM pH8, NaCl 150 mM, DTT 1 mM. Cells were lysed mechanically by using lysing matrix E (silica and glass beads) and in a FastPrep-24TM 5G (MP BiomedicalsTM).

## WCR strain rearing and larval diet

Western corn rootworm (WCR, *Diabrotica virgifera virgifera*, Coleoptera: Chrysomelidae, EPPO code DIABVI) were mass-reared following the procedures of [76–78]. A non-diapause WCR colony (USDA ARS, Bookings, USA) was used to infest artificial diet-based bioassays with neonates (<24h old) (see procedures below). The population is considered susceptible to most insecticides or novel agents as it had not been exposed to any of those, and therefore resistance is considered unlikely [79, 80]. Insect colonies resistant to Cry3Bb1 and Gpp34Ab1/Tpp35Ab1 are described in Ludwick et al. (2018). As described, these insect strains initially evolved some level of resistance in the field, were crossed to a non-diapausing strain, and thereafter were continuously selected on seeds expressing Cry3Bb1 or Gpp34Ab1/Tpp35Ab1 in USDA-ARS facilities in Columbia, MO USA.

For susceptible strains, two weeks prior the bioassays, soil dishes with freshly laid eggs had been removed from WCR adult rearing cages to allow sufficient incubation time until egg hatch. Eggs were washed with cool tap water with <0.01% NaOCl through a 300 μm mesh sieve. Around 5000 eggs were transferred to sterilized, slightly moist river sand (< 200 μm grains) in Petri dishes. They were incubated at 24 ± 2˚C in darkness for 8 to 12 days until hatching started. One the day before a bioassay, the ready-to-hatch eggs were again washed and sieved. Eggs were then again placed on sterile moist sand onto slightly moist tissue paper

into a dish to allow clean hatching conditions of new neonates and their use for bioassays. For resistant strains, the methods of Ludwick et al. [81] and Huynh et al. [82] were used.

WCR diet was prepared one day prior to the bioassay. The diet was prepared under semi-sterile conditions following methods described previously [82–84]. For the susceptible WCR stains, a commercial southern corn rootworm diet (Frontier #F9800B, Frontier Scientific Ltd., USA) plus additions of maize root powder and food colour were used to make the WCR diet. To prepare 100 ml of diet, 13.8 g of the #F9800B diet was ground and added to 88 ml fluid agar consisting of 1.5 g agar (CAS 9002-18-0, Chejeter, Japan) and deionized water. After blending and cooling to 55°C to 60°C, 0.75 g ground lyophilized maize roots (GLH5939 Pioneer, USA, or Phileaxx RAGT, Hungary) were added into the diet mixture, followed by the addition of 0.1 g green food colour for better larval observation (Les Artistes, France). Then, 1.7 to 1.8 ml 10% (w/v) KOH were added to reach a pH between 6.2 and 6.5. This mix was blended again, and then stirred at 50°C to 55°C. Then, 190 μl diet solution was pipetted into each 330 μl well of 96-well plates filling each to around 2/3$^{rd}$. Then, the 96-well plates with diet were allowed to dry in a laminar flow cabinet for 45 minutes, and then stored at 3°C to 5°C overnight. Diets for work with resistant strains followed Man et al. (2019).

## Insect bioassays *in vitro*

WCR bioassays on susceptible strains were conducted in 96-well plates by adding 20 μL of a treatment to each well. Each treatment was applied to 8 wells of each of six plates per bioassay (N = 48). Each treatment-dose combination was tested in at least two biological replicates. Order of treatments were shifted every other plate to avoid edge effects. Plates were dried for 1 to 1.5 h, and then cooled for 1 h in a fridge at 3°C to 5°C. One neonate larva was placed per well using a fine artist brush. A fast-moving, healthy-looking larva was chosen, and lifted from the end of its abdomen with the brush, moved towards a well surface, and allowed to crawl off the brush onto the diet. Larvae were not placed in treatment column order but rectangular to avoid systemic errors. After every 12th larva, the brush was cleaned in 70% ethanol followed by sterile tap water. The filled plate was closed with an optically clear adhesive qPCR seal sheet (#AB-1170, Thermo Scientific, USA or #BS3017000, Bioleader, USA) allowing data assessments without opening the plate. Four to five holes were made with flamed 00-insect pins into the seal per well to allow aeration. The plates were incubated at 24 ± 2°C and 50 to 70% relative humidity (RH) in the dark in a ventilated incubator (Friocell 22, MMM Medcenter, Munich, Germany) for 5 days. A bacterial lysate containing Gpp34Ab1/Tpp35Ab1 (https://www.rcsb.org/) was used as a positive control. Cleared empty lysate, which was a bacterial lysate produced in the same conditions and from the same bacterial strain containing the same expression vector as the other lysates but without the gene encoding the pesticidal protein, served as the negative control, as was sterilized water. Insecticidal activity was scored as dead, stunted (survived with no growth), no activity (growth equal to controls) after 3 and 5 days. Feeding was assessed through observing food remains, frass, and diet in the larval gut to assure that diet and a treatment had been ingested.

Statistical analyses were determined using a Fisher exact test comparing each sample result to the negative control. Results were considered positive if larval mortality and stunting were statistically higher than in the negative control and recorded at 3- and 5-days post treatment. Stunting was qualitatively assessed as an indicator for sublethal effects in comparison to the size and form of larvae in the negative controls.

WCR bioassays on resistant strains (Gpp34Ab1/Tpp35Ab1 and Cry3Bb1) were conducted with similar methods as for susceptible WCR strain except for the following points. The diet used in the bioassays was WCRMO2 [82]. The experiment consisted of a series of 3 plates

totaling 3 x 8 wells (column) (3 replicates per sample) and 24 larvae per treatment and controls (N = 24). Five days after treatment, mortality was assessed, and experimental plates were placed in a -20˚C freezer (Thermo Fisher Scientific TSE 400A, Marietta, GA) overnight. The next day the plates were removed and thawed to room temperature and the total weights of surviving larvae from each treatment was recorded in a Sartorius micro scale (model # MSE6-6S-000-DM, Sartorius Corporate, Göttingen, Germany) and then divided by the number of individuals to calculate mean larval weight per 8-well replicate. One-way analysis of variance (ANOVA) and Tukey's HSD post-hoc test were used to assess effects of treatments as well as differences among the treatments.

Lepidopteran bioassays were conducted in 96-well plates by adding 20 μL of crude lysate to 200 μL diet (General Purpose Lepidoptera Diet Frontier Scientific F9772 diet for ECB, FAW and CEW, and Southwestern Corn Borer Diet, Frontier Scientific F0635 plus Vitamin Mix, Frontier Scientific F0717 and Neomycin Sulfate, Sigma N1876). The diet was dispensed into a 96-well plate using an 8-channel pipette (Finnpipette Multistepper 4540500 Thermo Fisher, Marietta, GA), delivering 200 μl into each well and allowed to cool. Then, 20 μl of lysate was dispensed (Socorex Acura 826 XS 20–200 μl, Ecublens, Switzerland) onto each well of the diet plate as directed in the experimental plan. The wells were dried in a laminar flow hood for approximately 1 hour or until all inoculated wells were dried. All lepidopteran insects used in these experiments were received as eggs. All lepidopteran eggs were hatched in a 29˚C, 60% RH, 0:24L:D growth chamber (Conviron A1000, Manitoba, Canada). Eight wells per replicate and 3 replicates per treatment were infested with a single larva for each experiment providing 24 datapoints per experiment for each sample. After infesting each plate 2 holes were poked into the sealing film using a 000-insect pin and plates were placed in the growth chamber (Conviron A1000, Manitoba, Canada) at 26˚C, 50% RH, 0:24 L:D for 5 days. Insecticidal activity was scored as dead, stunted (survived with no growth), no activity (growth similar to negative controls) after 5 days. After scoring, the experimental plates were placed in a -80˚C freezer (Thermo Fisher Scientific TSE 400A, Marietta, GA) overnight. The next day the plates were removed and thawed to room temperature and the total weights (A&D GR-200 Analytical Balance, Columbia, MD) for each sample replicate were collected and then divided by the number of individuals to calculate mean larval weight per 8-well replicate. Statistical analyses were determined using a Fisher exact test comparing each sample results to the negative control.

All information on the origin of the insect strains is described in S3 Table.

## Supporting information

**S1 Fig. Multiple amino acid sequence alignment of component 1 of PIP45, PIP75 and GDI protein families.** All sequence information is presented in the S1 Table. GDI proteins are from this study, PIP sequences are from the patent WO2016/114973.
(TIF)

**S2 Fig. Multiple amino acid sequence alignment of component 2 of PIP45, PIP75 and GDI protein families.** All sequence information is presented in the S1 Table. GDI proteins are from this study. PIP sequences are from patent WO2016/114973.
(TIF)

**S3 Fig. Activity of homologs of GDI0005A and GDI0006A tested as pairs against western corn rootworm larvae.** Phylogenetic tree was built with the Neighbor-joining method using Jukes-Cantor Genetic Distance model (Geneious Tree Builder from Geneious).
(TIF)

**S4 Fig. Activity of GDI0005A and GDI0006A and commercial toxins Gpp34Aa1/ Tpp35Ab1 and Cry3Bb1 on western corn rootworm strains susceptible and resistant to commercial Bt toxins Gpp34Aa1/Tpp35Ab1 and Cry3Bb1. (A)**, **(B)**, and **(D)** describe the mortality and **(C)** describes the larval weight 5 days after treatment in diet overlay bioassays. (TIF)

**S1 Table. Information and sequences of all pesticidal proteins described in the paper.** (XLSX)

**S2 Table. GDI0005A/GDI0006A lysate activity against lepidopteran maize pests using diet overlay bioassays.** ECB: European corn borer *Ostrinia nubilalis*, FAW: fall armyworm *Spodoptera frugiperda*; SWCB: southwestern corn borer *Diatraea grandiosella*; CEW: corn earworm *Helicoverpa zea*. Fisher exact test was used to compare each treatment to the negative control at significance level P < 0.05. (XLSX)

**S3 Table. Information on the origin of the insect strains studied.** (XLSX)

**S1 Raw image.** (JPG)

## Acknowledgments

We express our sincere thanks to Limagrain and Genective colleagues for their contributions to this study and especially Jérôme Martin, David Comeau, Jorge Duarte, Hervé Duborjal, Gregory DesmaizieresChristian Faye. This article is dedicated to the memory of Dr. Cécile Persillon who was highly implicated in this project and passed away during the time of the project in June 2020. Cécile Persillon was an employee of Protéus, a subsidiary of the Seqens company.

We would like to acknowledge Anne-Nathalie Volkoff (INRAe, DGMI, Montpellier) and Collin Berry for reviewing the manuscript. Our thanks also go to Abedelaziz Heddi (head of the BF2I Laboratory, UMR INRA/INSA of Lyon) for his critical advice on the project during its implementation.

## Author Contributions

**Conceptualization:** Virginie Guyon, David Vallenet, Wyatt Paul, Thierry Heulin, Stefan Toepfer, Christophe Sallaud.

**Data curation:** Rania Jabeur, Szabolcs Toth, Adriano E. Pereira, Man P. Huynh, Erin Boland, Mickael Bosio, Pete Clark, David Vallenet.

**Formal analysis:** Rania Jabeur, Szabolcs Toth, Adriano E. Pereira, Man P. Huynh, Pete Clark, Bruce E. Hibbard, Stefan Toepfer.

**Funding acquisition:** Thierry Heulin, Christophe Sallaud.

**Investigation:** Virginie Guyon, Szabolcs Toth, Adriano E. Pereira, Man P. Huynh, Erin Boland, Laurent Beuf, Pete Clark, Wafa Achouak, Carine Audiffrin, Wyatt Paul, Stefan Toepfer.

**Methodology:** Adriano E. Pereira, Man P. Huynh, Erin Boland, Laurent Beuf, Pete Clark, Carine Audiffrin, Wyatt Paul, Stefan Toepfer.

**Project administration:** François Torney, Christophe Sallaud.

**Resources:** Zakia Selmani, David Vallenet, Wafa Achouak, Carine Audiffrin, François Torney, Thierry Heulin, Stefan Toepfer.

**Supervision:** Virginie Guyon, Wafa Achouak, François Torney, Wyatt Paul, Thierry Heulin, Bruce E. Hibbard, Stefan Toepfer, Christophe Sallaud.

**Validation:** Laurent Beuf, Pete Clark, Carine Audiffrin, François Torney, Wyatt Paul, Stefan Toepfer.

**Visualization:** Rania Jabeur, Mickael Bosio.

**Writing – original draft:** Rania Jabeur.

**Writing – review & editing:** Virginie Guyon, Szabolcs Toth, Adriano E. Pereira, Man P. Huynh, Wafa Achouak, François Torney, Wyatt Paul, Thierry Heulin, Bruce E. Hibbard, Stefan Toepfer, Christophe Sallaud.

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
