## [Decision Letter · Decision Letter 0]

4 Jan 2022

PONE-D-21-31910A novel binary pesticidal protein from Chryseobacterium arthrosphaerae controls Diabrotica virgifera virgifera via a different mode of action to existing commercial pesticidal proteinsPLOS ONE

Dear Dr. Sallaud,

Thank you for submitting your manuscript to PLOS ONE. After careful consideration, we feel that it has merit but does not fully meet PLOS ONE’s publication criteria as it currently stands. Therefore, we invite you to submit a revised version of the manuscript that addresses the points raised during the review process.

We look forward to receiving your revised manuscript.

Kind regards,

Sengottayan Senthil-Nathan, Ph D

Academic Editor

PLOS ONE

Journal Requirements:

The project was supported by the SEMAE “French Interprofessional Organization for Seeds and Plants” former GNIS association under the call named “Fond Diabrotica”. 

RJ, VG, ST, AP, AMH, ZS, EB, MB, LB, PC, DV, WA, CA, FT, WP, TH, BH, ST, CS 

Grant number = “Fond Diabrotica" 

Funder = French Interprofessional Organization for Seeds and Plants” former GNIS association under the call named

URL of the funder = https://www.gnis.fr/

Additional Editor Comments:

Minor revision is requested

Points to be consider:

In brief: A complete check for formatting and typographic errors must be done. I can’t understand several points because of language barrier.

Specific points:

Abstract needs to be modified to deliver a proper meaning.

Introduction

A clear picture of resistance and advantage of bio-insecticide should be provided.

At least refer following documents (10.3389/fphys.2019.01591, 10.1016/j.envint.2017.12.038, 10.3390/molecules26123695, 10.1016/j.aquatox.2020.105474)

I’m not convinced with method. Please prove detailed methodology for each experiment.

A proper explanation of obtained results with appropriate citations indicating all the results shall be included in the discussion

Complete update of citations throughout from introduction to discussion is needed.

Reviewers' comments:

Reviewer's Responses to Questions

**Comments to the Author**

1. Is the manuscript technically sound, and do the data support the conclusions?

Reviewer #1: Yes

Reviewer #2: Yes

2. Has the statistical analysis been performed appropriately and rigorously? 

Reviewer #1: Yes

Reviewer #2: Yes

3. Have the authors made all data underlying the findings in their manuscript fully available?

Reviewer #1: Yes

Reviewer #2: Yes

4. Is the manuscript presented in an intelligible fashion and written in standard English?

Reviewer #1: Yes

Reviewer #2: Yes

5. Review Comments to the Author

Reviewer #1: In this study, the effect of GDI0005A and GDI0006A recombinant proteins on the Western Corn Rootworm (WCR) was investigated. The results showed that these proteins showed good insecticidal activity against the larvae of WCR. This reviewers thinks that the experiment is well design and properly executed . The manuscript is well written, especially for a person without a strong background in molecular biology. Hence it can reach general readers of PlosOne .This reviewer believes, this manuscript it fits for PlosOne after some minor revisions (below).

1. Brief introduction why this strain was selected could be included in the introduction.

2. Despite the method is clear, it is not clear how and why the binary toxins (GDI0005A and GDI0006A) were detected at the first place as candidate of the novel toxin? From the genome?

3. Predicted protein sequences were annotated based on homology to sequences in public databases. Provide details what public databases that are used in this study.

4. Reference for the bioinformatics tools (not only links to their gihub) i.e. SPADES and Prodigal will be appreciated.

5. Why after scoring, the experimental plates were placed in a -80˚C freezer prior calculating the total weights?

Reviewer #2: The manuscript entitled A novel binary pesticidal protein from Chryseobacterium arthrosphaerae controls

Diabrotica virgifera virgifera via a different mode of action to existing commercial

pesticidal proteins..”is a result of planned research. In general, the paper is well written, concise and the subject is interesting.

Manuscript contains significant information to justify publication. Some minor corrections need to be performed to improve the paper.

Comments:

1. Proof read the entire manuscript for typographical errors and fix all grammatical errors. In case extensive revision of the English language has been recommended.

2. The manuscript title is not particularly appropriate and needs to be revised in terms of grammar.

3. Please revise the abstract. The abstract should contain a summary of the results and should exclude the irrelevant content. The abstract must provide enough information that will allow a reader to understand the key findings of the research.

4. Latest references are missing in the entire manuscript; following are few latest references suggested, can be used in material method and discussion section.

5. Introduction section should be clear with research gaps and how the present work going to contribute in advanced scientific research

6. Author can cite the related articles on pyrethroids such as doi: 10.1007/s13205-016-0372-3;DOI: 10.3390/microorganisms8020223;https://doi.org/10.1007/s13205-016-0541-4;doi.org/10.3389/fmicb.2019.01778

6. PLOS authors have the option to publish the peer review history of their article (what does this mean?). If published, this will include your full peer review and any attached files.

Reviewer #1: No

Reviewer #2: **Yes: **Pankaj Bhatt

---

## [Author Response · Author response to Decision Letter 0]

4 Mar 2022

Responses to Editor Comments:

Authors: The manuscript has been reviewed according to PLOS ONE’s style requirements. 

Authors: Funding information was removed from the manuscript and the amended statements are found in this cover letter 

Authors: GDI0005A and GDI0006A are available in Genbank under accession numbers OK428600 and OK428601 respectively. Information has been added in the supplemental table S1. They have also been registered in Bacterial Pesticidal Protein Resource Center [BPPRC] database under the name Xpp84Aa1 (GDI0005A) and Xpp85Aa1 (GDI0006A). This information has been added in the text of the manuscript in the first chapter of the result part. 

 4. PLOS ONE now requires that authors provide the original uncropped and unadjusted images underlying all blot or gel results reported in a submission’s figures or Supporting Information files. 

Authors: An original image of the Figure 8B has been uploaded to Plos site. 

5. Please review your reference list to ensure that it is complete and correct

Authors: The reference list has been fully reviewed 

Additional Editor Comments: Minor revision is requested

In brief: A complete check for formatting and typographic errors must be done. I can’t understand several points because of language barrier. 

Authors: The manuscript has been thoroughly re-cheeked for formatting and typographic errors. It has also been reviewed by another English native speakers. 

Specific points: 

Abstract needs to be modified to deliver a proper meaning. 

Authors: We have entirely revised the abstract and hope it has largely improved.

Introduction

A clear picture of resistance and advantage of bio-insecticide should be provided. 

At least refer following documents (10.3389/fphys.2019.01591, 10.1016/j.envint.2017.12.038, 10.3390/molecules26123695, 10.1016/j.aquatox.2020.105474)

Authors: Two suggested references (ref 36 & 37) were included in the introduction to add a better overview on bio-insecticide advantages 

Method: I’m not convinced with method. Please prove detailed methodology for each experiment 

Authors: All methods have been reviewed and improved by all co-authors. Please kindly refer to our improvements visible through the tracked changes.

A proper explanation of obtained results with appropriate citations indicating all the results shall be included in the discussion 

Authors: The discussion has been reviewed to better explain all results obtained with appropriate citations. 

Complete update of citations throughout from introduction to discussion is needed. 

Authors: we have reviewed all citations throughout all the document

Reviewers' comments:

Reviewer #1: 

In this study, the effect of GDI0005A and GDI0006A recombinant proteins on the Western Corn Rootworm (WCR) was investigated. The results showed that these proteins showed good insecticidal activity against the larvae of WCR. This reviewer thinks that the experiment is well design and properly executed. The manuscript is well written, especially for a person without a strong background in molecular biology. Hence it can reach general readers of PlosOne. This reviewer believes, this manuscript it fits for PlosOne after some minor revisions (below).

Authors: We are pleased to hear that the reviewer sees high value in our scientific findings and believes our experiments were done correctly and our manuscript is well written. We like to also thank for the valuable comments we have now addressed.

Authors: We have revised the manuscript regarding English and readability. The paper has been reviewed by an English native speaker. We have also double-checked for formatting and typing errors

1. Brief introduction why this strain was selected could be included in the introduction. 

Authors: A description on the strategy used to select the strain has been added 

2. Despite the method is clear, it is not clear how and why the binary toxins (GDI0005A and GDI0006A) were detected at the first place as candidate of the novel toxin? From the genome? 

Authors: Yes, the candidates were detected from the genome - The information has been added more clearly in the first paragraph of the results in association with the description of the strategy mentioned in the previous paragraph 1.) 

3. Predicted protein sequences were annotated based on homology to sequences in public databases. Provide details what public databases that are used in this study. 

Authors: The information on which public databases was used is now mentioned in the text. It is BPPRC, UniProtKB, public patent databases such as EPO and USPTO

4. Reference for the bioinformatics tools (not only links to their gihub) i.e. SPADES and Prodigal will be appreciated. 

Authors: References for SPADES and Prodigal were added

5. Why after scoring, the experimental plates were placed in a -80˚C freezer prior calculating the total weights?

Authors: Larvae cannot be weighed while they are still alive, so it is why they are killed by placing them in a freezer. 

Reviewer #2: The manuscript entitled A novel binary pesticidal protein from Chryseobacterium arthrosphaerae controls 

Diabrotica virgifera virgifera via a different mode of action to existing commercial

pesticidal proteins..”is a result of planned research. In general, the paper is well written, concise and the subject is interesting. Manuscript contains significant information to justify publication. Some minor corrections need to be performed to improve the paper.

Authors: We are pleased to hear that the reviewer finds our scientific findings interesting and believes our manuscript is concise and well written. We like to also thank for the valuable comments we have now addressed.

1. Proof read the entire manuscript for typographical errors and fix all grammatical errors. In case extensive revision of the English language has been recommended. 

Authors: The manuscript has been thoroughly re-cheeked for formatting and typographic errors. It has also been reviewed by another English native speakers. Please refer to all the tracked changes in the manuscript.

2. The manuscript title is not particularly appropriate and needs to be revised in terms of grammar. 

Authors: A novel title is proposed 

3. Please revise the abstract. The abstract should contain a summary of the results and should exclude the irrelevant content. The abstract must provide enough information that will allow a reader to understand the key findings of the research. 

Authors: The abstract has been reviewed by several authors. Please refer to all the tracked changes in the manuscript.

4. Latest references are missing in the entire manuscript; following are few latest references suggested, can be used in material method and discussion section. 

Authors: All references were reviewed 

5. Introduction section should be clear with research gaps and how the present work going to contribute in advanced scientific research 

Authors: The introduction has been reviewed to better explain research gaps and how the present work can contribute in advanced scientific research. 

6. Author can cite the related articles on pyrethroids such as doi: 10.1007/s13205-016-0372-3;DOI: 10.3390/microorganisms8020223;https://doi.org/10.1007/s13205-016-0541-4;doi.org/10.3389/fmicb.2019.01778

Authors: The following reference was added in the introduction – 

DOI: 10.3390/microorganisms8020223;

---

## [Decision Letter · Decision Letter 1]

5 Apr 2022

A novel binary pesticidal protein from Chryseobacterium arthrosphaerae controls western corn rootworm by a different mode of action to existing commercial pesticidal proteins

PONE-D-21-31910R1

Dear Dr. Sallaud,

We’re pleased to inform you that your manuscript has been judged scientifically suitable for publication and will be formally accepted for publication once it meets all outstanding technical requirements.

Kind regards,

Sengottayan Senthil-Nathan, Ph D

Academic Editor

PLOS ONE

Additional Editor Comments (optional):

The manuscript can be accepted now
---

## [Editor Report · Acceptance letter]

28 Apr 2022

PONE-D-21-31910R1 

A novel binary pesticidal protein from *Chryseobacterium arthrosphaerae* controls western corn rootworm by a different mode of action to existing commercial pesticidal proteins

Dear Dr. Sallaud:

I'm pleased to inform you that your manuscript has been deemed suitable for publication in PLOS ONE. Congratulations! Your manuscript is now with our production department. 

Kind regards, 

on behalf of

Prof. Sengottayan Senthil-Nathan 

Academic Editor

PLOS ONE